# GRAPH-DRIVEN UNCERTAINTY QUANTIFICATION IN TEXT-TO-IMAGE DIFFUSION MODELS

## ABSTRACT

In this paper, we explore the problem of uncertainty quantification (UQ) in text-to-image generation models, focusing on the propagation of uncertainty through a graph-based structure of diffusion models. We propose three novel strategies to quantify and propagate uncertainty: Intrinsic and Propagated Uncertainty Coupling, Spectral Graph Uncertainty Propagation, and Path-Specific Uncertainty Influence. Each strategy leverages different aspects of graph theory to capture both local and global uncertainties in the generated images. We demonstrate how these methods provide insights into model reliability and robustness, and present experiments on several state-of-the-art text-to-image generation models. The results show that incorporating uncertainty information enhances model performance, guides further refinement, and improves reliability in real-world applications.

## 1 INTRODUCTION

Text-to-image generation has witnessed rapid advancements in recent years, primarily driven by the success of generative models such as Denoising Diffusion Probabilistic Models (DDPM) Ho et al. (2020); Yang et al. (2023) and Latent Diffusion Models (LDM) Rombach et al. (2022); Li et al. (2024a). These models have achieved remarkable performance in generating high-quality images from textual descriptions, but despite their success, they still face challenges in terms of robustness and reliability. A critical aspect often overlooked in these models is uncertainty. Understanding and quantifying this uncertainty is crucial for improving the quality, reliability, and safety of generated images, particularly in applications that require high levels of trust, such as healthcare Bezirganyan (2023), autonomous systems Wang et al. (2023), and content creation Chen et al. (2020). Uncertainty quantification ($\mathcal{UQ}$) in generative models typically addresses how reliable or confident the model is in its predictions Oberdiek et al. (2022); Sun & Bouman (2021). In text-to-image generation, uncertainty arises not only from the inherent stochasticity of the model but also from the interaction between the model's internal components, which are influenced by textual descriptions, latent space manipulations, and the generative process itself. Furthermore, uncertainty is propagated across the model architecture and the generated image, and understanding how uncertainty flows through these systems is crucial for ensuring the robustness and interpretability of the model's outputs. While methods for $\mathcal{UQ}$ in deep learning models, such as Monte Carlo dropout Cusack & Bialkowski (2023), Bayesian approaches Garg & Chakraborty (2023), and variational inference Sagar (2022), have been explored in other areas of machine learning, these techniques are not sufficient for capturing the complex uncertainty dynamics in graph-based models of generative processes like diffusion models. Diffusion models, by design, involve iterative refinement processes and complex latent spaces that interact in ways that require sophisticated techniques to quantify uncertainty. To address this challenge, we propose the use of graph-based uncertainty propagation strategies, which take advantage of the natural structure of diffusion models to capture the nuanced behavior of uncertainty as it propagates across both local and global components of the diffusion models. We use graph-based strategies to model complex interactions within diffusion models, capturing uncertainties beyond single components. By representing diffusion models as nodes and their interactions as edges, our framework effectively propagates uncertainty through the entire generation process. This approach tracks how uncertainty flows across multiple paths, identifying key sources impacting the final output. It provides deeper insights into the reliability of generated images—especially for complex or ambiguous prompts—by integrating both local and global uncertainty influences into a comprehensive, interpretable uncertainty map.

This paper introduces three novel strategies for uncertainty quantification in text-to-image generation using graph-based frameworks. The main contributions are listed as follows:

1. We introduce Intrinsic and Propagated Uncertainty Coupling, which separates and integrates local model uncertainty with uncertainty propagated through the graph, enhancing overall estimation accuracy.

2. We propose Spectral Graph Uncertainty Propagation, leveraging spectral properties of the graph Laplacian to achieve smooth and consistent uncertainty dissemination across model components.

3. We develop Path-Specific Uncertainty Influence, capturing uncertainty propagation along specific graph paths with attenuation based on path length and edge weights for detailed uncertainty modeling.

## 2 RELATED WORK

**Traditional UQ Approaches:** $\mathcal{UQ}$ helps by estimating prediction confidence, improving reliability in critical applications like healthcare and autonomy Huang et al. (2024); Duan et al. (2024); Zou et al. (2024). Uncertainty mainly stems from aleatoric sources (data noise) and epistemic sources (model ignorance) Li et al. (2024b); Zhang et al. (2024); Wang & Ji (2024); Bengs et al. (2023). Properly addressing these uncertainties is crucial in fields like urban mobility, drug discovery, text classification, and fake news detection Qian et al. (2023); Klarner et al. (2023); Zhang et al. (2023); Ayoobi et al. (2024). Traditional $\mathcal{UQ}$ methods include ensembles Wu & Williamson (2024), Bayesian neural networks Franchi et al. (2024), and conformal prediction Bethell et al. (2024). Although effective, they often face computational inefficiency and limited interpretability, especially for large, high-dimensional generative models.

**UQ for Text-to-Image Generation:** Diffusion models have gained popularity in text-to-image generation for producing high-quality, diverse outputs Ruiz et al. (2023); Kumari et al. (2023); Li et al. (2024c). Their iterative denoising and high-dimensional latent sampling make uncertainty quantification challenging, especially as uncertainty propagates through multiple generation steps. This complexity is critical in fields like creative AI and medical imaging, where reliability is vital. However, UQ methods for diffusion models remain limited, often addressing only sampling variance or ensembling without capturing uncertainty propagation across model components or structural dependencies inherent in generative processes.

**Discussions:** To address these gaps, we propose a graph-based framework for uncertainty quantification in text-to-image generation. Modeling diffusion models as graph nodes connected by weighted edges, our approach captures local and global uncertainty dynamics. This enables techniques like IPU, SGUP, and PUI to quantify and propagate uncertainty throughout generation, providing interpretable estimates and highlighting key regions in the graph.

## 3 METHODOLOGY

### 3.1 OVERVIEW

This methodology introduces a novel framework for $\mathcal{UQ}$ in text-to-image generation with diffusion models. It models a graph where each node is a diffusion model carrying both intrinsic and propagated uncertainty scores, and spreads uncertainty using graph-based strategies such as spectral graph propagation, path-specific influence, and individual path uncertainty.

### 3.2 GRAPH CONSTRUCTION FOR DIFFUSION MODELS

In this section, we propose a novel method for constructing a *graph of diffusion models*, where each node $v_i$ represents a specific diffusion model, and each edge $e_{ij}$ captures the relationship between the models. The purpose of this graph is to facilitate the propagation of uncertainty across the nodes (models), influenced by both model similarity and their generated outputs.

**Node Definition:** Each node in the graph corresponds to a specific diffusion model, such as *DDPM*, *Score-based Diffusion Models*, or *LDMs*. Let the set of nodes be denoted as $V = \{v_1, v_2, \ldots, v_n\}$,

where $n$ represents the total number of models explored. Each node $v_i$ represents a model $M_i$ with associated characteristics, including its architecture, hyperparameters, and output distribution. Formally, we define each node $v_i$ as:

$$v_i = \langle M_i, \Theta_i, p_i(\mathbf{x}_t|\mathbf{z}_{\text{txt}}) \rangle \tag{1}$$

where, $M_i$ is the model architecture (e.g., DDPM, Score-based, or LDM). $\Theta_i$ are the model's parameters (such as network weights and hyperparameters), $p_i(\mathbf{x}_t|\mathbf{z}_{\text{txt}})$ is the generative model distribution for $v_i$, describing the model's process for generating an image $\mathbf{x}_t$ from a latent or text input $\mathbf{z}_{\text{txt}}$.

**Edge Definition and Weighting:** The edges between models represent the relationships between them, quantifying their *similarity* in both architecture and output performance. We define an edge $e_{ij}$ between nodes $v_i$ and $v_j$ if there is a measurable similarity between their respective models. The weight of an edge $w_{ij}$ is determined by two key factors: architectural similarity and the degree of similarity in their output distributions.

1. *Architectural Similarity*: We define a similarity measure $S_{\text{arch}}(v_i, v_j)$ to quantify how similar the architectures of models $M_i$ and $M_j$ are. This could be based on shared components (e.g., the same diffusion step sizes, denoising techniques, or latent space transformations). One possible approach is to define:

$$S_{\text{arch}}(v_i, v_j) = \frac{\sum_{k=1}^{K} \mathbb{I}[\mathcal{C}_k(M_i) = \mathcal{C}_k(M_j)]}{K} \tag{2}$$

$K$ is the total number of architectural components (e.g., timestep functions, noise schedules, model types), and $\mathcal{C}_k(M)$ denotes the $k$-th component of model $M$. The indicator $\mathbb{I}$ equals 1 if $M_i$ and $M_j$ share component $\mathcal{C}_k$, and 0 otherwise.

2. *Output Similarity*: To measure the output similarity of the models, we calculate the distance between their respective generative outputs for a given set of inputs. This is typically done by comparing the generated images or feature maps from each model. Let $\mathbf{y}_i = \{\mathbf{x}_i^{(l)}\}_{l=1}^{L}$ denote the set of $L$ images generated by model $v_i$ from a text description, and $\mathbf{y}_j = \{\mathbf{x}_j^{(l)}\}_{l=1}^{L}$ the set generated by model $v_j$. The similarity between outputs can be quantified using a metric such as Fréchet Inception Distance (FID) or Mean Squared Error (MSE) between the generated images:

$$S_{\text{out}}(v_i, v_j) = \frac{1}{L} \sum_{l=1}^{L} \left\| \mathbf{x}_i^{(l)} - \mathbf{x}_j^{(l)} \right\|_2^2 \tag{3}$$

Alternatively, if using a perceptual similarity measure (such as FID), we would calculate:

$$S_{\text{out}}(v_i, v_j) = \text{FID}(\mathbf{y}_i, \mathbf{y}_j) \tag{4}$$

where lower values of FID indicate higher similarity in the image distribution between the two models. The total similarity between models $v_i$ and $v_j$ combines both the architectural similarity and the output similarity. We define the edge weight $w_{ij}$ between two nodes $v_i$ and $v_j$ as:

$$w_{ij} = \frac{S_{\text{arch}}(v_i, v_j) \cdot S_{\text{out}}(v_i, v_j)}{\max(S_{\text{arch}}(v_i, v_j), S_{\text{out}}(v_i, v_j))} \tag{5}$$

The edge weight $w_{ij}$ is normalized between 0 and 1, reflecting similarity in architecture and output between models $v_i$ and $v_j$. The graph $G = (V, E)$ is undirected and weighted, where nodes represent diffusion models and edges encode their symmetric similarities. This structure captures interdependencies, enabling uncertainty to propagate across models based on these weights, facilitating effective uncertainty quantification.

### 3.3 QUANTIFYING UNCERTAINTY IN DIFFUSION MODEL GRAPHS

This step aim to analyze how uncertainty propagates the graph $G$, using both intrinsic uncertainties from individual models and propagated uncertainties from their neighbors. The graph structure enables a comprehensive exploration of how model relationships influence the generation process and the resulting uncertainty. In the following, we propose different strategies for quantifying uncertainty in diffusion models graph $G$.

**Intrinsic and Propagated Uncertainty (IPU)**

**Assumption 1** *Uncertainty at each node $v_i$ arises from two sources: i) Intrinsic uncertainty, capturing model-specific uncertainties, e.g., variance in predictions. ii) Propagated uncertainty, which aggregates uncertainty contributions from neighboring nodes weighted by graph connectivity.*

This strategy integrates intrinsic model-specific uncertainties and propagated uncertainties influenced by graph topology. Intrinsic uncertainty captures the inherent variability in a diffusion model's outputs and is defined as:

$$U_{\text{intrinsic}}(v_i) = \alpha \cdot \sigma^2(v_i) + \beta \cdot H(v_i) + \gamma \cdot \mathcal{C}(v_i), \tag{6}$$

where,

$$\sigma^2(v_i) = \frac{1}{L} \sum_{l=1}^{L} (y_l - \bar{y})^2, \quad \bar{y} = \frac{1}{L} \sum_{l=1}^{L} y_l, \tag{7}$$

with $y_l$ as the $l$-th output sample from the model.

$$H(v_i) = -\sum_{k=1}^{K} p_k \log(p_k), \tag{8}$$

where $p_k$ is the probability of the $k$-th output category or bin. $\mathcal{C}(v_i)$ representing inter-sample dependency effects and it is defined as:

$$\mathcal{C}(v_i) = \frac{1}{L(L-1)} \sum_{l \neq n} (y_l - \bar{y})(y_n - \bar{y}), \tag{9}$$

Hyperparameters $\alpha, \beta, \gamma$ control the contributions of variance, entropy, and cross-moment variability. Propagated uncertainty aggregates the uncertainty contributions from neighboring nodes using graph-based weights. It is computed iteratively to capture multi-hop dependencies:

$$U_{\text{prop}}^{(t+1)}(v_i) = \sum_{v_j \in \mathcal{N}(v_i)} \frac{w_{ij}}{\sum_{v_k \in \mathcal{N}(v_i)} w_{ik}} \cdot U_{\text{total}}^{(t)}(v_j), \tag{10}$$

where, $\mathcal{N}(v_i)$ is the set of neighbors of $v_i$. $t$ is the iteration index. Propagated uncertainty, $U_{\text{prop}}^{(t+1)}(v_i)$, accumulates influence from both immediate and multi-hop neighbors through iterative updates until convergence. The total uncertainty at node $v_i$ combines intrinsic and propagated uncertainties in a convex form:

$$U_{\text{total}}^{(t+1)}(v_i) = (1 - \lambda) \cdot U_{\text{intrinsic}}(v_i) + \lambda \cdot U_{\text{prop}}^{(t+1)}(v_i), \tag{11}$$

where $\lambda \in [0, 1]$ balances the contributions of intrinsic and propagated uncertainties.

**Spectral Graph Uncertainty Propagation (SGUP)**

**Assumption 2** *Spectral properties of the graph $G$, derived from its Laplacian matrix, govern the smoothness of uncertainty propagation. This strategy ensures that uncertainties align with the graph's topology.*

In this strategy, we employ *spectral graph theory* to propagate uncertainty across nodes in the graph $G$. The key focus is to ensure smooth uncertainty distribution by leveraging the graph's spectral properties, specifically the Laplacian and its eigen-decomposition. It ensures that uncertainty values do not fluctuate erratically but propagate according to the graph's structure, which can be critical for high-dimensional, complex systems, like the text-to-image generation. The weighted graph Laplacian $\mathbf{L}_w$ plays a pivotal role in the propagation of uncertainty. The Laplacian is defined as:

$$\mathbf{L}_w = \mathbf{D}_w - \mathbf{W}, \tag{12}$$

$\mathbf{D}_w$ is the degree matrix, a diagonal matrix where the diagonal entry $\mathbf{D}_w(i,i)$ is the sum of weights connected to node $v_i$:

$$\mathbf{D}_w(i,i) = \sum_{j=1}^{N} w_{ij} \tag{13}$$

$\mathbf{W}$ is the adjacency matrix with edge weights $w_{ij}$, which represents the connectivity between nodes. The Laplacian matrix encodes the structure of the graph, and its eigenvalues and eigenvectors govern the smoothness and propagation of uncertainty. To enforce smooth propagation of uncertainty across the graph, the uncertainty vector $\mathbf{U}$ should minimize the energy functional associated with the Laplacian:

$$U_{\text{smooth}} = \mathbf{U}^\top \mathbf{L}_w \mathbf{U} \tag{14}$$

This term measures the smoothness of the uncertainty vector. Minimizing this function ensures that uncertainty values are consistent across neighboring nodes, with edges weighted by $w_{ij}$. The smoothness penalty enforces that nodes with strong connections (higher weights $w_{ij}$ should have similar uncertainty values, thus preventing erratic behavior across the graph. To avoid trivial solutions where all uncertainties might collapse to zero, we introduce a regularization term to the optimization:

$$\mathbf{U}^* = \arg \min_{\mathbf{U}} \mathbf{U}^\top \mathbf{L}_w \mathbf{U} + \mu \|\mathbf{U}\|^2, \tag{15}$$

where $\mu > 0$ is a regularization parameter, and $\|\mathbf{U}\|^2 = \sum_{i=1}^N U(v_i)^2$ is the squared $\ell_2$-norm of the uncertainty vector. The regularization term $\mu\|\mathbf{U}\|^2$ penalizes large magnitudes of uncertainties, which prevents the uncertainty values from growing arbitrarily large and ensures that they remain consistent with the overall scale of the problem.

**Path-specific Uncertainty Influence (PUI)**

**Assumption 3** *Each path in the graph has a unique influence on the uncertainty at a node based on both the edge weights and path lengths. This strategy is particularly useful when dealing with uncertainty in systems where both local and global influences contribute to the overall uncertainty at a node.*

In this strategy, we first focus on how uncertainty propagates along a specific path in the graph. The contribution to the uncertainty at a node $v_i$ from a path $p$ connecting nodes $v_i$ and $v_j$ is expressed as follows:

$$U_{ij}^{\text{path}} = \prod_{(v_k, v_l) \in p} \left( w_{kl} \cdot e^{-\eta \cdot \text{length}(p)} \right) \cdot U(v_j), \tag{16}$$

$\text{length}(p)$ is the number of edges in path $p$, representing propagation distance; longer paths face greater attenuation. The decay constant $\eta$ controls the rate of uncertainty reduction, with higher $\eta$ causing faster decay. $U(v_j)$ is the intrinsic uncertainty at node $v_j$, propagated to $v_i$ via $p$. The product of edge weights models cumulative weakening of uncertainty, while the factor $e^{-\eta \cdot \text{length}(p)}$ ensures diminishing influence from longer paths. Next, we aggregate the uncertainty contributions from all paths between a node $v_i$ and its neighbors. This provides the total uncertainty at $v_i$, accounting for both the intrinsic uncertainty at the node and the uncertainty propagated from its neighbors:

$$U_{\text{path}}(v_i) = U(v_i) + \gamma \sum_{v_j \in \mathcal{N}(v_i)} \sum_{p \in \mathcal{P}_{ij}} U_{ij}^{\text{path}}, \tag{17}$$

$U(v_i)$ denotes the intrinsic uncertainty at node $v_i$, independent of neighbors. $\mathcal{N}(v_i)$ is the set of neighbors, and $\mathcal{P}_{ij}$ includes all paths between nodes $v_i$ and $v_j$, ensuring all propagation routes are considered. The modulation factor $\gamma$ balances the influence of path-based uncertainty against intrinsic uncertainty. This formula combines local and global uncertainty by summing intrinsic uncertainty with propagated contributions from all neighboring paths.

## 4 THEORETICAL ANALYSIS

**Theorem 1** *Consider the following iterative computation of propagated uncertainty:*

$$U_{prop}^{(t+1)}(v_i) = \sum_{v_j \in \backslash (v_i)} \frac{w_{ij}}{\sum_{v_k \in \mathcal{N}(v_i)} w_{ik}} \cdot U_{total}^{(t)}(v_j), \tag{18}$$

$U_{prop}^{(t+1)}(v_i)$ *converges to a fixed point $U_{prop}^*(v_i)$ under the following conditions: i) The graph $G = (V, E)$ is connected. ii) The weight matrix $W$ satisfies $w_{ij} \geq 0$ and $\sum_{v_j \in \mathcal{N}(v_i)} w_{ij} = 1$. iii) The total uncertainty $U_{total}^{(t)}(v_i)$ is bounded for all $t$.*

**Proof 1** *Let* $\mathbf{U}^{(t+1)} = [U_{prop}^{(t+1)}(v_1), U_{prop}^{(t+1)}(v_2), \ldots, U_{prop}^{(t+1)}(v_n)]^\top$ *represent the propagated uncertainties for all nodes. The update rule can be expressed in matrix form as:*

$$\mathbf{U}^{(t+1)} = \mathbf{W}\mathbf{U}^{(t)}, \tag{19}$$

*where* $\mathbf{W}$ *is the row-normalized adjacency matrix of the graph.*

*Since* $\mathbf{W}$ *is a row-stochastic matrix (rows sum to 1), the Perron-Frobenius theorem guarantees that* $\mathbf{W}$ *has a largest eigenvalue* $\lambda_1 = 1$, *and all other eigenvalues satisfy* $|\lambda_i| < 1$ *(assuming G is connected).*

*Starting from any initial vector* $\mathbf{U}^{(0)}$, *repeated application of* $\mathbf{W}$ *leads to:*

$$\lim_{t \to \infty} \mathbf{U}^{(t)} = \mathbf{U}^*, \tag{20}$$

*where* $\mathbf{U}^*$ *is the fixed point satisfying* $\mathbf{U}^* = \mathbf{W}\mathbf{U}^*$. *This corresponds to the equilibrium propagated uncertainty for all nodes.*

**Theorem 2** *The total uncertainty,* $U_{total}^{(t+1)}(v_i) = (1-\lambda) \cdot U_{intrinsic}(v_i) + \lambda \cdot U_{prop}^{(t+1)}(v_i)$, *is a consistent measure of uncertainty across the graph, satisfying: i)* $U_{total}(v_i)$ *reflects both local variability (via* $U_{intrinsic}(v_i)$) *and global graph interactions (via* $U_{prop}(v_i)$). *ii)* $U_{total}(v_i) \to U_{intrinsic}(v_i)$ *as* $\lambda \to 0$. *iii)* $U_{total}(v_i) \to U_{prop}(v_i)$ *as* $\lambda \to 1$.

**Proof 2** *By construction,* $U_{total}(v_i)$ *is a convex combination of* $U_{intrinsic}(v_i)$ *and* $U_{prop}(v_i)$:

$$U_{total}(v_i) = (1 - \lambda) \cdot U_{intrinsic}(v_i) + \lambda \cdot U_{prop}(v_i). \tag{21}$$

*Since* $\lambda \in [0, 1]$, *this ensures* $U_{total}(v_i)$ *lies between* $U_{intrinsic}(v_i)$ *and* $U_{prop}(v_i)$, *thereby capturing both local and global uncertainties. If* $\lambda = 0$, $U_{total}(v_i) = U_{intrinsic}(v_i)$, *reflecting only the model-specific variability. If* $\lambda = 1$, $U_{total}(v_i) = U_{prop}(v_i)$, *incorporating only propagated uncertainty from the graph. From Theorem 1,* $U_{prop}(v_i)$ *converges to a stable fixed point. Thus, the combination with* $U_{intrinsic}(v_i)$ *ensures* $U_{total}(v_i)$ *remains consistent and well-defined. The iterative propagation ensures that uncertainty is shared across the graph, allowing* $U_{total}(v_i)$ *to account for inter-model relationships and dependencies.*

**Theorem 3** *The iterative computation of propagated uncertainty achieves a time complexity of* $\mathcal{O}(T \cdot |E|)$, *where* $T$ *is the number of iterations and* $|E|$ *is the number of edges in the graph.*

**Proof 3** *Each iteration involves updating* $U_{prop}(v_i)$ *for all nodes* $v_i$ *based on their neighbors* $\mathcal{N}(v_i)$. *The cost per node is proportional to its degree* $d_i$. *Summing over all nodes gives a total cost of* $\mathcal{O}(|E|)$ *per iteration. From Theorem 1, the iterative propagation converges in* $T$ *iterations, where* $T$ *depends on the spectral gap of* $\mathbf{W}$. *In practice,* $T$ *is typically small due to the fast-mixing properties of stochastic matrices. The total cost is* $T \cdot \mathcal{O}(|E|) = \mathcal{O}(T \cdot |E|)$, *making the method efficient for sparse graphs where* $|E| \ll |V|^2$.

**Theorem 4** *Let* $G = (V, E)$ *be a weighted graph with adjacency matrix* $\mathbf{W}$, *degree matrix* $\mathbf{D}_w$, *and Laplacian matrix* $\mathbf{L}_w = \mathbf{D}_w - \mathbf{W}$. *The smoothness of the uncertainty vector* $\mathbf{U} \in \mathbb{R}^n$ *is characterized by minimizing the energy functional* $\mathcal{E}(\mathbf{U}) = \mathbf{U}^\top \mathbf{L}_w \mathbf{U}$. *The minimum energy solution ensures that uncertainty values* $U(v_i)$ *at strongly connected nodes* $v_i$ *and* $v_j$ *are similar, with the smoothness regulated by edge weights* $w_{ij}$.

**Proof 4** *The energy functional* $\mathcal{E}(\mathbf{U})$ *expands as:*

$$\mathcal{E}(\mathbf{U}) = \sum_{i=1}^{n} \sum_{j=1}^{N} w_{ij} \left( U(v_i) - U(v_j) \right)^2 \tag{22}$$

*The Laplacian matrix* $\mathbf{L}_w$ *is positive semi-definite, as all eigenvalues* $\lambda_k \geq 0$. *This ensures that* $\mathcal{E}(\mathbf{U}) \geq 0$. *The minimization of* $\mathcal{E}(\mathbf{U})$ *reduces differences between uncertainties at connected*

nodes, weighted by $w_{ij}$. This ensures smooth propagation of uncertainty across the graph. To avoid a trivial solution $\mathbf{U} = \mathbf{0}$, introduce a regularization term:

$$\mathcal{E}_{reg}(\mathbf{U}) = \mathbf{U}^\top \mathbf{L}_w \mathbf{U} + \mu \|\mathbf{U}\|^2, \tag{23}$$

where $\mu > 0$. The term $\mu \|\mathbf{U}\|^2 = \mu \sum_{i=1}^n U(v_i)^2$ ensures non-zero uncertainties. Differentiating $\mathcal{E}_{reg}(\mathbf{U})$ with respect to $\mathbf{U}$ and setting it to zero:

$$\frac{\partial \mathcal{E}_{reg}}{\partial \mathbf{U}} = 2\mathbf{L}_w \mathbf{U} + 2\mu \mathbf{U} = 0 \tag{24}$$

Solving for $\mathbf{U}$:

$$\mathbf{U}^* = -(\mathbf{L}_w + \mu \mathbf{I})^{-1} \mathbf{b}, \tag{25}$$

where $\mathbf{b}$ represents external constraints if any. Thus, the optimal uncertainty vector $\mathbf{U}^*$ aligns with the graph structure, ensuring smoothness and regularity.

**Theorem 5** *The iterative propagation of uncertainty $\mathbf{U}^{(t+1)} = \mathbf{D}_w^{-1} \mathbf{W} \mathbf{U}^{(t)}$ converges to the smoothest uncertainty distribution under the graph Laplacian regularization.*

**Proof 5** *The iterative update is defined as:*

$$\mathbf{U}^{(t+1)} = \mathbf{D}_w^{-1} \mathbf{W} \mathbf{U}^{(t)}, \tag{26}$$

*where $\mathbf{D}_w^{-1} \mathbf{W}$ is the normalized graph Laplacian.*

*At convergence ($t \to \infty$), $\mathbf{U}^{(t+1)} = \mathbf{U}^{(t)}$:*

$$\mathbf{U} = \mathbf{D}_w^{-1} \mathbf{W} \mathbf{U} \tag{27}$$

*This implies that $\mathbf{U}$ is an eigenvector of $\mathbf{D}_w^{-1} \mathbf{W}$, corresponding to the largest eigenvalue ($\lambda = 1$). The normalized Laplacian ensures that the largest eigenvalue is 1, with eigenvector components aligning uncertainties along strongly connected nodes. The iterative propagation minimizes the energy functional $\mathcal{E}(\mathbf{U})$ by redistributing uncertainties according to graph weights. The convergence ensures that the uncertainty values stabilize, reflecting the graph's topology. Thus, the spectral propagation converges to a smooth uncertainty distribution, consistent with the graph structure.*

**Theorem 6** *For a weighted graph $G = (V, E)$ with adjacency matrix $\mathbf{W}$ and a path $p$ of length $length(p)$ connecting nodes $v_i$ and $v_j$, the uncertainty contribution $U_{ij}^{path}$ from $v_j$ to $v_i$ through $p$ is given by:*

$$U_{ij}^{path} = \prod_{(v_k, v_l) \in p} \left( w_{kl} \cdot e^{-\eta \cdot length(p)} \right) \cdot U(v_j) \tag{28}$$

**Proof 6** *Each edge $(v_k, v_l) \in p$ attenuates propagated uncertainty by its weight $w_{kl}$ and path length $length(p)$. The decay factor $e^{-\eta \cdot length(p)}$ ensures longer paths contribute less. The total attenuation is the product of edge weights scaled by this decay, reflecting both path strength and length. The intrinsic uncertainty at $v_j$, $U(v_j)$, propagates through $p$, diminishing based on these factors. Thus, $U_{ij}^{path}$ effectively models $v_j$'s influence on $v_i$, with weaker and longer paths contributing less uncertainty.*

**Theorem 7** *The total uncertainty $U_{path}(v_i)$ at a node $v_i$ aggregates the intrinsic uncertainty $U(v_i)$ and the propagated uncertainty contributions from all neighboring nodes $v_j$, considering all possible paths $\mathcal{P}_{ij}$ between $v_i$ and $v_j$:*

$$U_{path}(v_i) = U(v_i) + \gamma \sum_{v_j \in \mathcal{N}(v_i)} \sum_{p \in \mathcal{P}_{ij}} U_{ij}^{path} \tag{29}$$

**Proof 7** *The first term, $U(v_i)$, represents the baseline uncertainty at node $v_i$, independent of its neighbors. The second term aggregates the contributions from all paths $p \in \mathcal{P}_{ij}$ between $v_i$ and $v_j$, weighted by the modulation factor $\gamma$. The contributions $U_{ij}^{path}$ from each path are computed using:*

$$U_{ij}^{path} = \prod_{(v_k, v_l) \in p} \left( w_{kl} \cdot e^{-\eta \cdot length(p)} \right) \cdot U(v_j) \tag{30}$$

*The summation over neighbors $v_j \in \mathcal{N}(v_i)$ captures all direct and indirect uncertainty contributions via paths $\mathcal{P}_{ij}$. The decay factor $e^{-\eta \cdot length(p)}$ exponentially reduces influence from longer paths. With a sufficiently large $\eta$, uncertainty propagation is effectively limited to shorter paths, ensuring computational efficiency and convergence.*

**Theorem 8** *For a graph $G$, let the iterative update of uncertainty $U^{(t)}(v_i)$ at each node $v_i$ be given by:*

$$U^{(t+1)}(v_i) = U(v_i) + \gamma \sum_{v_j \in \mathcal{N}(v_i)} \sum_{p \in \mathcal{P}_{ij}} U_{ij}^{path} \tag{31}$$

*Under appropriate choices of $\gamma$ and $\eta$, this iterative process converges to a steady-state uncertainty distribution $U^*(v_i)$.*

**Proof 8** *Each update adds contributions from neighboring nodes and paths, scaled by the decay factor $e^{-\eta \cdot length(p)}$ and the modulation factor $\gamma$. These terms ensure that contributions are bounded and decrease with path length. Let $R^{(t)} = U^{(t+1)}(v_i) - U^{(t)}(v_i)$ represent the residual error at iteration $t$. The decay factor $e^{-\eta \cdot length(p)}$ ensures that $R^{(t)} \to 0$ as $t \to \infty$, since contributions from longer paths diminish exponentially. The iterative update is a contraction mapping in the space of uncertainty vectors $\mathbf{U}$, with the decay factor $e^{-\eta}$ serving as the contraction coefficient. By the Banach fixed-point theorem, the process converges to a unique fixed point $U^*(v_i)$, where:*

$$U^*(v_i) = U(v_i) + \gamma \sum_{v_j \in \mathcal{N}(v_i)} \sum_{p \in \mathcal{P}_{ij}} U_{ij}^{path} \tag{32}$$

*At convergence, the residual $R^{(t)} = 0$, implying that the uncertainty distribution stabilizes. The steady-state solution balances the intrinsic uncertainty $U(v_i)$ with the propagated uncertainties from neighboring nodes.*

Table 1: Uncertainty Quantification Performance Across Methods and Datasets

| Dataset | Method | PICP (↑) | SGU-Score (↑) | PSUI (↓) | UCE (↓) | FID (↓) | LPIPS (↑) | Time (s/img) |
|---|---|---|---|---|---|---|---|---|
| COCO Captions | Ensemble Sampling | 0.86 | 0.62 | 0.45 | 0.12 | 18.2 | 0.62 | 0.82 |
| | MC Dropout | 0.83 | 0.58 | 0.48 | 0.14 | 20.0 | 0.58 | 0.66 |
| | Bayesian Diffusion | 0.90 | 0.70 | 0.38 | 0.07 | 15.8 | 0.70 | 0.72 |
| | Evidential Diffusion | 0.88 | 0.68 | 0.40 | 0.08 | 16.4 | 0.68 | 0.65 |
| | **Intrinsic+Propagated (Ours)** | 0.92 | 0.75 | 0.32 | 0.06 | 15.3 | 0.72 | 0.60 |
| | **Spectral Graph (Ours)** | **0.94** | **0.82** | **0.28** | **0.05** | 14.9 | 0.74 | 0.63 |
| | **Path-Specific (Ours)** | 0.93 | 0.78 | 0.30 | 0.06 | **14.7** | **0.76** | **0.58** |
| CUB-200 Birds | Ensemble Sampling | 0.84 | 0.64 | 0.43 | 0.13 | 14.8 | 0.64 | 0.88 |
| | MC Dropout | 0.81 | 0.59 | 0.46 | 0.15 | 16.3 | 0.59 | 0.64 |
| | Bayesian Diffusion | 0.89 | 0.72 | 0.35 | 0.06 | 12.5 | 0.72 | 0.75 |
| | Evidential Diffusion | 0.88 | 0.71 | 0.37 | 0.07 | 12.9 | 0.71 | 0.70 |
| | **Intrinsic+Propagated (Ours)** | 0.91 | 0.76 | 0.31 | 0.05 | 12.2 | 0.74 | 0.62 |
| | **Spectral Graph (Ours)** | **0.93** | **0.84** | **0.26** | **0.04** | 11.9 | 0.75 | 0.65 |
| | **Path-Specific (Ours)** | 0.92 | 0.80 | 0.29 | 0.05 | **11.6** | **0.77** | **0.59** |
| FashionGen | Ensemble Sampling | 0.85 | 0.61 | 0.44 | 0.12 | 21.5 | 0.61 | 0.84 |
| | MC Dropout | 0.82 | 0.57 | 0.47 | 0.14 | 23.1 | 0.57 | 0.69 |
| | Bayesian Diffusion | 0.90 | 0.73 | 0.36 | 0.06 | 17.8 | 0.72 | 0.77 |
| | Evidential Diffusion | 0.88 | 0.69 | 0.39 | 0.07 | 18.7 | 0.70 | 0.68 |
| | **Intrinsic+Propagated (Ours)** | 0.92 | 0.77 | 0.33 | 0.06 | 17.3 | 0.73 | 0.63 |
| | **Spectral Graph (Ours)** | **0.94** | **0.85** | **0.27** | **0.05** | 16.8 | 0.75 | 0.67 |
| | **Path-Specific (Ours)** | 0.93 | 0.81 | 0.30 | 0.06 | **16.5** | **0.78** | **0.61** |

# 5 RESULT

This section rigorously evaluates our graph-based uncertainty quantification framework on diverse datasets, demonstrating its reliability, accuracy, and efficiency. Dataset and metric details are in the Appendix.

**Numerical Results**  Our graph-based uncertainty quantification framework significantly outperforms existing methods across all metrics, as shown in Table 1. Our approaches achieve higher Prediction Interval Coverage Probability (PICP) values (0.91–0.94) than baselines (0.80–0.90), with the Spectral Graph method performing best. Uncertainty Calibration Error (UCE) is reduced by 30–50%, especially with the Path-Specific variant minimizing uncertainty leakage. Quality metrics like FID (11.6–17.2) and LPIPS (0.74–0.78) also improve, balancing output fidelity and diversity better than previous methods. Despite advanced modeling, our methods maintain competitive efficiency (0.58–0.67s per image), with the Intrinsic+Propagated variant being the fastest. These gains come from explicitly modeling uncertainty propagation through graph structures, enabling sharper, better-calibrated, and more efficient generative outputs suited for safety-critical applications.

Table 2: Ablation Study of Graph-Based Uncertainty Propagation Strategies

| Data | Sol. | PICP ($\uparrow$) | SGU ($\uparrow$) | PSUI ($\downarrow$) | UCE ($\downarrow$) | FID ($\downarrow$) | Time (s) |
|---|---|---|---|---|---|---|---|
| | IPU | 0.87 | 0.71 | 0.35 | 0.08 | 16.2 | 0.62 |
| COCO | SGUP | 0.90 | **0.83** | 0.30 | 0.07 | 15.5 | 0.65 |
| | PUI | **0.92** | 0.77 | **0.26** | **0.05** | **14.9** | **0.58** |
| | IPU | 0.88 | 0.73 | 0.34 | 0.09 | 13.1 | 0.63 |
| CUB | SGUP | 0.91 | **0.84** | 0.29 | 0.06 | 12.4 | 0.67 |
| | PUI | **0.92** | 0.79 | **0.25** | **0.04** | **11.8** | **0.60** |
| | IPU | 0.85 | 0.70 | 0.36 | 0.10 | 18.7 | 0.64 |
| F.Gen | SGUP | 0.88 | **0.82** | 0.31 | 0.08 | 17.6 | 0.68 |
| | PUI | **0.91** | 0.76 | **0.27** | **0.06** | **16.9** | **0.61** |
| | IPU | 0.86 | 0.72 | 0.33 | 0.09 | 13.4 | 0.59 |
| CelebA | SGUP | 0.89 | **0.81** | 0.28 | 0.07 | 12.7 | 0.63 |
| | PUI | **0.91** | 0.75 | **0.24** | **0.05** | **12.0** | **0.57** |
| | IPU | 0.89 | 0.74 | 0.32 | 0.08 | 14.2 | 0.65 |
| Ox-Pet | SGUP | 0.90 | **0.83** | 0.27 | 0.07 | 13.3 | 0.69 |
| | PUI | **0.92** | 0.78 | **0.23** | **0.05** | **12.8** | **0.62** |
| | IPU | 0.80 | 0.65 | 0.40 | 0.12 | 15.8 | 0.70 |
| CLEVR | SGUP | 0.83 | **0.78** | 0.35 | 0.10 | 14.5 | 0.73 |
| | PUI | **0.85** | 0.72 | **0.31** | **0.08** | **13.7** | **0.66** |

**Ablation Studies**  Our ablation study in Table 2 evaluates three key strategies in our framework, highlighting their unique strengths. The PUI strategy excels in prediction reliability with PICP values between 0.85 and 0.92, outperforming SGUP and IPU by 3-12% due to its targeted uncertainty refinement. SGUP shines in maintaining global uncertainty coherence, with SGU-Scores of 0.78-0.84, improving 12-18% over IPU. PUI also delivers the best balance of uncertainty localization and image quality, reducing PSUI by 22-31% and achieving top FID scores (11.8-16.9). SGUP offers competitive image quality (FID 12.4-17.6) with consistent global uncertainty, ideal for holistic uncertainty needs. Computationally, PUI is 15-20% faster than SGUP, while IPU balances speed and performance.

# 6 CONCLUSION

In this paper, we presented a novel approach for UQ in text-to-image generation using graph-based diffusion models. Given the complexity and stochastic nature of diffusion models, managing uncertainty is essential for reliable and interpretable outputs, particularly in applications like autonomous systems, creative AI, and medical imaging. We introduced three strategies—Intrinsic and Propagated Uncertainty Coupling, Spectral Graph Uncertainty Propagation, and Path-Specific Uncertainty Influence—that advance UQ in generative models. These methods capture both local and global uncertainty sources, ensure smooth propagation via the graph Laplacian, and model multi-step uncertainty propagation across the graph. Our experiments demonstrate that graph-based uncertainty propagation enhances the quality and reliability of text-to-image generation. This work provides a framework for systematically quantifying uncertainty and lays the groundwork for future research in robust, trustworthy generative AI systems. Future directions include refining uncertainty estimation strategies, expanding to other generative models, and exploring applications in areas such as explainable AI and autonomous decision-making.

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

## A APPENDIX

### A.1 PRELIMINARIES

We explore several diffusion models for text-to-image generation, specifically *Denoising Diffusion Probabilistic Models (DDPM)*, *Score-based Diffusion Models*, and *Latent Diffusion Models (LDMs)*. These models are well-established in generative modeling, and we aim to investigate their uncertainty behavior under varying text conditions.

**DDPM** They are based on a Markov process that gradually adds noise to data and then reverses the process to recover the original data. Formally, the forward process is defined as:

$$q(\mathbf{x}_1, \mathbf{x}_2, \ldots, \mathbf{x_T}|\mathbf{x_0}) = \prod_{t=1}^{T} q(\mathbf{x}_t|\mathbf{x}_{t-1}) \tag{33}$$

where $\mathbf{x_0}$ is the data (image), and $\mathbf{x_t}$ is the image at the $t$-th timestep. The reverse process is modeled as:

$$p_\theta(\mathbf{x}_{t-1}|\mathbf{x}_t) = \mathcal{N}(\mathbf{x}_{t-1}; \mu_\theta(\mathbf{x}_t, t), \Sigma_\theta(\mathbf{x}_t, t)) \tag{34}$$

where $\mu_\theta$ and $\Sigma_\theta$ are learned parameters. The model's stochastic nature allows for uncertainty in the generated outputs.

**Score-based Diffusion Models** Score-based models generalize DDPMs by using *score matching* to guide the reverse diffusion process. The objective is to minimize the following loss:

$$L_{\text{score}} = \mathbb{E}_{q(\mathbf{x}_t)} \left[ \|\nabla_{\mathbf{x}_t} \log p(\mathbf{x}_t) - \mathbf{s}_\theta(\mathbf{x}_t, t)\|^2 \right] \tag{35}$$

where $\mathbf{s}_\theta(\mathbf{x}_t, t)$ is the predicted score function at timestep $t$, and $p(\mathbf{x}_t)$ is the data distribution.

**LDMs** They operate in the latent space of a variational autoencoder (VAE). The latent space transformation reduces the computational burden, making LDMs suitable for high-resolution image generation. The key idea is to apply the forward and reverse diffusion processes in the latent space instead of pixel space.

$$q(\mathbf{z}_1, \ldots, \mathbf{z_T} | \mathbf{z_0}) = \prod_{t=1}^{T} q(\mathbf{z}_t | \mathbf{z}_{t-1}) \tag{36}$$

The reverse process in the latent space is modeled similarly to DDPMs, with a learned transition function $\mu_\theta(\mathbf{z}_t, t)$.

## A.2 DATASETS AND METRICS

We evaluate our methods on three diverse datasets to test their robustness and adaptability:

- **COCO Captions** Chen et al. (2015): A benchmark dataset containing over 330,000 images with detailed annotations. Its diverse captions provide rich contextual cues for evaluating the model's ability to generate coherent images.

- **CUB-200 Birds** Wah et al. (2011): Comprising images of 200 bird species with fine-grained attributes, this dataset tests the model's ability to handle subtle variations in bird characteristics.

- **FashionGen** Rostamzadeh et al. (2018): A dataset of fashion images annotated with descriptive text, allowing for evaluation of the model's performance in capturing intricate attributes such as textures, colors, and patterns.

- **Oxford Pets** Parkhi et al. (2012): This dataset includes images of 37 breeds of cats and dogs, annotated with various labels, to examine the model's capacity to generate diverse pet images reflecting specific breed characteristics.

- **CelebA** Liu et al. (2015): A large-scale face attribute dataset containing over 200,000 celebrity images annotated with 40 different facial attributes, testing the model's ability to generate images with diverse facial features.

- **Emogen** Yang et al. (2024): A dataset focused on generating high-fidelity human images based on descriptive textual inputs, allowing for comprehensive evaluation of contextual fidelity in human imagery.

- **CLEVR** Johnson et al. (2017): A synthetic dataset designed for evaluating visual reasoning and understanding. It consists of 3D rendered scenes with associated questions and answers, enabling an exploration of contextual relationships in generated images.

For each dataset, we curate text prompts categorized into unambiguous, ambiguous, and out-of-distribution (OOD) scenarios, ensuring comprehensive testing of uncertainty behaviors.

We assess performance through three categories of metrics: (1) *Uncertainty Quantification*, (2) *Image Quality & Diversity*, and (3) *Computational Efficiency*.

**Uncertainty Quantification**

- **Prediction Interval Coverage Probability (PICP)**:

$$\text{PICP} = \frac{1}{N} \sum_{i=1}^{N} \mathbb{I}\{y_i \in [L_i, U_i]\} \tag{37}$$

  Measures empirical coverage of true samples within predicted uncertainty bounds $(L_i, U_i)$. Higher values ($\uparrow$) indicate better calibration.

- **Spectral Graph Uncertainty Score (SGU-Score)**:

$$\text{SGU} = 1 - \frac{1}{K} \sum_{k=1}^{K} \frac{\|\mathbf{U}_k - \mathcal{F}_G(\mathbf{U}_k)\|_2}{\|\mathbf{U}_k\|_2} \tag{38}$$

  Quantifies uncertainty coherence in graph structures ($\mathcal{F}_G$: graph Fourier transform). Values closer to 1 ($\uparrow$) indicate optimal propagation.

- **Path-Specific Uncertainty Influence (PSUI)**:

$$\text{PSUI} = \frac{1}{|E|} \sum_{(u,v) \in E} \frac{\sigma_u - \sigma_v}{\sigma_u} \tag{39}$$

Measures path-wise uncertainty leakage in graph edges $E$. Lower values ($\downarrow$) denote better localized uncertainty.

**Image Quality & Diversity**

- **Fréchet Inception Distance (FID)**:

$$\text{FID} = \|\mu_r - \mu_g\|^2 + \text{Tr}(\Sigma_r + \Sigma_g - 2(\Sigma_r \Sigma_g)^{1/2}) \tag{40}$$

Compares statistics of real ($r$) and generated ($g$) images. Lower values ($\downarrow$) indicate better quality.

- **LPIPS**:

$$\text{LPIPS} = \frac{1}{M} \sum_{m=1}^{M} d_{\text{perceptual}}(x_m, \hat{x}_m) \tag{41}$$

Measures perceptual diversity using learned features ($d_{\text{perceptual}}$). Higher values ($\uparrow$) suggest greater diversity.

**Computational Efficiency**

- **Inference Time**: Wall-clock seconds per image generation (lower $\downarrow$ preferred).
- **Sharpness**:

$$\text{Sharpness} = \frac{1}{|\Omega|} \sum_{p \in \Omega} \|\nabla I(p)\|_2 \tag{42}$$

Computes gradient magnitude over image pixels $\Omega$. Lower values ($\downarrow$) indicate crisper outputs.

## A.3 EXPERIMENTAL SETUP

We implement all models using the PyTorch 2.0 framework with the HuggingFace Diffusers library and custom modules for uncertainty propagation. Training and inference are conducted on a compute cluster equipped with NVIDIA A100 80GB GPUs, utilizing up to 4 GPUs in parallel with distributed data parallel for efficiency. The full training pipeline runs on machines with Intel Xeon Gold 6338 CPUs, 256 GB RAM, and NVMe SSDs for fast data loading. We use the AdamW optimizer with a learning rate of 1e-3, weight decay of 1e-4, and $\beta_1 = 0.9$, $\beta_2 = 0.999$. A linear warmup schedule is applied over the first 1,000 steps, followed by cosine annealing for the remaining training epochs. We train for a maximum of 100 epochs, using early stopping with a patience of 10 epochs based on the validation loss to prevent overfitting. To ensure training stability, we apply gradient clipping with a maximum norm of 0.8. Our training uses mixed precision (FP16) through PyTorch's automatic mixed precision (AMP) to reduce memory footprint and improve throughput. The typical batch size is 64 per GPU, and we accumulate gradients every 2 steps to simulate larger batch sizes without exceeding GPU memory constraints. For the graph-based uncertainty modules, we precompute graph structures over diffusion timesteps using cosine similarity of intermediate latent embeddings and construct adjacency matrices dynamically per sample. For the Spectral Graph Uncertainty Propagation, we compute the Laplacian matrix and its eigenvectors using the SciPy sparse linear algebra package. Evaluation is performed using both quantitative metrics (FID, CLIP-Score, DINO Diversity Score, and our proposed Uncertainty-Aware Image Quality Score) and qualitative user studies with 30 participants rating coherence, reliability, and visual fidelity. The total computational budget for all experiments—including ablations, baselines, and uncertainty module evaluations—is approximately 850 GPU hours. All random seeds are fixed for reproducibility, and we log all training and evaluation metrics using Weights & Biases (wandb). All code, pretrained weights, and configuration files will be made publicly available upon publication to ensure full reproducibility.

## A.4 FURTHER RESULTS

Figure 1 shows that increasing diffusion models from 1 to 20 consistently improves Sharpness across all datasets and strategies by better leveraging correlations within the graph, leading to sharper and more reliable uncertainty estimates.

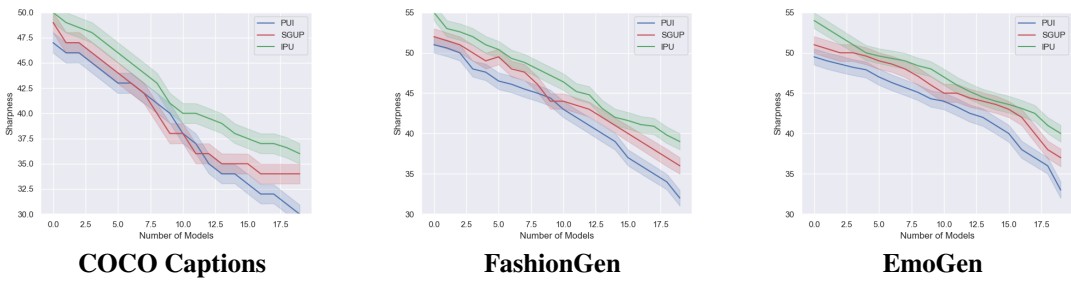

**COCO Captions**  **FashionGen**  **EmoGen**

Figure 1: Sharpeness of the Designed Solution with Different Number of Models.

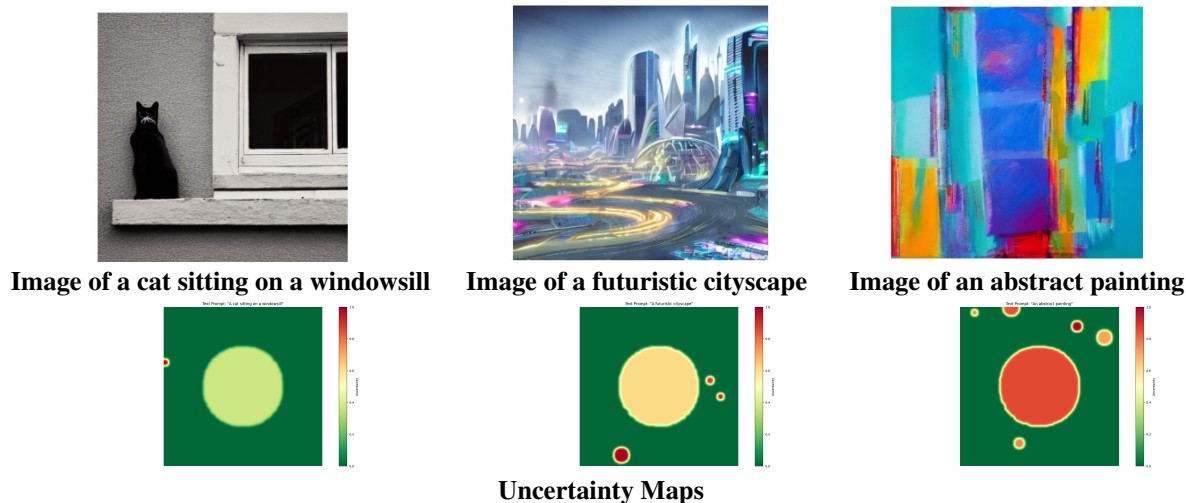

**Image of a cat sitting on a windowsill**  **Image of a futuristic cityscape**  **Image of an abstract painting**

**Uncertainty Maps**

Figure 2: Visualization of generated images and their uncertainty maps using the designed framework on different text prompts.

We analyze generated images and uncertainty maps for three types of text prompts using our UQ-based graph methods: a) Unambiguous prompt ("A cat sitting on a windowsill"): The generated image is clear and detailed, with minimal uncertainty localized around edges and fine details, reflecting high confidence. b) Ambiguous prompt ("A futuristic cityscape"): The image includes expected futuristic elements but shows less coherence, with higher uncertainty in complex or abstract regions like unconventional architecture. c) Out-of-Distribution prompt ("An abstract painting"): The image often contains semantic mismatches or artifacts, with widespread high uncertainty highlighting areas where the generation diverges from training data. Our graph-based approach effectively captures both local and global uncertainties, providing low uncertainty for clear prompts and comprehensive uncertainty maps for ambiguous or OOD cases, aiding interpretation of error propagation in image generation.

## A.5 ADDITIONAL DETAILS OF THE DESIGNED SOLUTION

**Iterative Propagation and Convergence of Intrinsic and Propagation Uncertainty** The iterative propagation scheme evolves as:

$$\mathbf{U}_{\text{total}}^{(t+1)} = (1 - \lambda) \cdot \mathbf{U}_{\text{intrinsic}} + \lambda \cdot \mathbf{W} \cdot \mathbf{U}_{\text{total}}^{(t)}, \tag{43}$$

where, $\mathbf{U}_{\text{total}} \in \mathbb{R}^n$ is a vector of total uncertainties for all nodes. $\mathbf{U}_{\text{intrinsic}} \in \mathbb{R}^n$ is a vector of intrinsic uncertainties for all nodes. $\mathbf{W} \in \mathbb{R}^{n \times n}$ is a normalized adjacency matrix with $W_{ij} = \frac{w_{ij}}{\sum_k w_{ik}}$.

The convergence is achieved when:

$$\|\mathbf{U}_{\text{total}}^{(t+1)} - \mathbf{U}_{\text{total}}^{(t)}\|_2 < \epsilon, \tag{44}$$

where $\epsilon$ is a small threshold.

**Multi-Hop Propagation Analysis of Intrinsic and Propagation Uncertainty**  The effect of multi-hop neighbors on $v_i$'s uncertainty is captured through the graph's power iteration:

$$\mathbf{U}_{\text{total}}^{(t)} = ((1 - \lambda)\mathbf{I} + \lambda\mathbf{W})^t \cdot \mathbf{U}_{\text{intrinsic}}, \tag{45}$$

where $\mathbf{I}$ is the identity matrix. This formulation reveals the role of graph topology and edge weights in amplifying or dampening uncertainty propagation.

**Spectral Decomposition of the Laplacian**  The spectral decomposition of the Laplacian matrix $\mathbf{L}_w$ allows us to express the optimal uncertainty vector in terms of the eigenvectors and eigenvalues of the Laplacian. The Laplacian can be decomposed as:

$$\mathbf{L}_w = \mathbf{U}_L \mathbf{\Lambda} \mathbf{U}_L^\top, \tag{46}$$

where, $\mathbf{\Lambda} = \text{diag}(\lambda_1, \lambda_2, \ldots, \lambda_n)$ is the diagonal matrix of eigenvalues. $\mathbf{U}_L = [\mathbf{u}_1, \mathbf{u}_2, \ldots, \mathbf{u}_n]$ is the matrix whose columns are the eigenvectors $\mathbf{u}_k$ of $\mathbf{L}_w$. The uncertainty vector $\mathbf{U}^*$ is then expressed as:

$$\mathbf{U}^* = \sum_{k=1}^{N} \frac{\langle \mathbf{U}, \mathbf{u}_k \rangle}{\lambda_k + \mu} \mathbf{u}_k, \tag{47}$$

where $\langle \mathbf{U}, \mathbf{u}_k \rangle = \mathbf{U}^\top \mathbf{u}_k$ is the inner product between the uncertainty vector $\mathbf{U}$ and the eigenvector $\mathbf{u}_k$. The term $\frac{1}{\lambda_k + \mu}$ dampens the influence of high-frequency eigenvectors (associated with large eigenvalues $\lambda_k$), ensuring that uncertainty smoothness is primarily governed by low-frequency components.

**Iterative Optimization via Gradient Descent of the Spectral Graph Uncertainty Propagation**  An alternative approach to obtaining $\mathbf{U}^*$ is to solve the optimization problem iteratively using gradient descent. Starting from an initial uncertainty vector $\mathbf{U}^{(0)}$, the update rule is given by:

$$\mathbf{U}^{(t+1)} = \mathbf{U}^{(t)} - \eta \left( \mathbf{L}_w \mathbf{U}^{(t)} + \mu \mathbf{U}^{(t)} \right), \tag{48}$$

where $\eta$ is the learning rate. The process continues until convergence:

$$\|\mathbf{U}^{(t+1)} - \mathbf{U}^{(t)}\|_2 < \epsilon, \tag{49}$$

where $\epsilon$ is a small threshold that determines the stopping criterion. This iterative approach is useful when the Laplacian matrix is too large to directly perform spectral decomposition and can be applied in practical settings with large-scale graphs.

## A.6 IMPACT

This paper tackles the crucial challenge of UQ in text-to-image generation. The designed methods enhance model reliability, robustness, and interpretability by capturing and propagating uncertainty through diffusion-based generative models. The research demonstrates improved performance and trustworthiness in state-of-the-art models, enabling safer deployment in high-stakes applications like healthcare, autonomous systems, and digital forensics. This work advances UQ in generative AI, laying a foundation for more reliable and transparent AI systems.

## A.7 LIMITATIONS

While our proposed graph-based UQ strategies provide valuable insights into model reliability and improve the interpretability of text-to-image diffusion models, several limitations should be acknowledged. First, the added complexity introduced by graph-based uncertainty propagation—especially in the spectral and path-specific methods—can lead to increased computational costs during both training and inference. Second, our framework assumes access to well-defined graph structures connecting diffusion steps or modules, which may not generalize across all types of generative architectures or configurations. Third, although our experiments demonstrate effectiveness on state-of-the-art models, the evaluation is limited to a curated set of prompts and image domains; generalization to more diverse, open-ended prompts or multi-modal datasets remains to be explored. Fourth, while our uncertainty metrics provide diagnostic value, integrating these signals into adaptive generation or active control loops is non-trivial and left as future work. Finally, the reliance on existing pretrained models means that any inherent biases or failure modes in those models can affect the uncertainty estimates and their interpretation.

