# OpenReview forum: "Graph-Driven Uncertainty Quantification in Text-to-Image Diffusion Models"
_ICLR.cc/2026/Conference — ICLR 2026 Conference Withdrawn Submission_

### Official Review · Reviewer_4aS2 · 2025-10-23

**Soundness:** 1
**Presentation:** 1
**Contribution:** 1
**Rating:** 2
**Confidence:** 4

**Summary:**

The authors introduce a novel graph-based framework to quantify uncertainty in text-to-image generation models. They model each diffusion model as a node in a graph and use strategies like Intrinsic-and-Propagated Uncertainty Coupling, Spectral Graph Uncertainty Propagation, and Path-Specific Uncertainty Influence to measure uncertainty. The method captures both local and global uncertainties.

**Strengths:**

1. The paper addresses an important and timely topic—uncertainty modeling in large generative models.

2. The paper proposes a novel approach with potentially valuable contributions to uncertainty estimation in diffusion-based architectures.

**Weaknesses:**

1. It is unclear how the metrics in Table 1 are computed. Are the authors training new models or evaluating pre-trained ones? Please clarify the experimental setup.

2. It is not clear how uncertainty contributes to improved model performance. Is uncertainty incorporated into the loss function during training or into the inference process? Typically, uncertainty in diffusion models is used for OOD detection or hallucination identification, so additional explanation is needed.

3. The paper introduces graphs of diffusion models but does not provide concrete examples. Each node is described as a diffusion model and edges represent similarity between models. Do such graphs exist in practice, or are they purely conceptual for this work?

4. Missing citations: Prior works [1, 2] already estimate uncertainty in diffusion models and should be cited for completeness.

5. Formatting issue: Several equations appear inline or too close to surrounding text (see Eqs. 11–16). These should be properly separated for readability.

6. The Results section lacks sufficient methodological detail. The paper discusses outcomes but does not explain how the results were obtained or what experimental setup produced them.


[1] Berry, Lucas, Axel Brando, and David Meger. "Shedding light on large generative networks: Estimating epistemic uncertainty in diffusion models." The 40th Conference on Uncertainty in Artificial Intelligence. 2024.

[2] Berry, Lucas, et al. "Seeing the Unseen: How EMoE Unveils Bias in Text-to-Image Diffusion Models." arXiv preprint arXiv:2505.13273 (2025).

**Questions:**

Please see weaknesses.

---

### Official Review · Reviewer_3EPp · 2025-10-31

**Soundness:** 1
**Presentation:** 1
**Contribution:** 2
**Rating:** 2
**Confidence:** 3

**Summary:**

The paper models a graph over diffusion models or components to propagate uncertainty and introduces three strategies for this: Intrinsic and Propagated Uncertainty coupling, Spectral Graph Uncertainty Propagation, and Path-Specific Uncertainty Influence. It provides convergence statements and reports improvements on several datasets.

**Strengths:**

The paper explores an area that is not theoretically well explored: uncertainty quantification in text-to-image diffusion models.

It uses concepts from graph theory to model the behavior of diffusion models.

**Weaknesses:**

1. The paper is at times ambiguous. How is the graph constructed for diffusion models? Is it built over a single model, or over latents during denoising? Both are mentioned in the text (L.103, L.758 say v_i is a diffusion model; L.747 says diffusion timesteps).

2. The related work section discusses UQ in broad diffusion contexts but omits prior work specific to T2I UQ (e.g., Towards Understanding and Quantifying Uncertainty for Text-to-Image Generation, Franchi et al.).

3. The definition of architectural similarity (L.126) seems over-simplified. Different architectural components generally do not have an equal effect.

4. L.138 is hard to justify as written. The term implicitly assumes a correspondence between the i-th image generated by each model. Also, L.133 mentions using the images themselves. in that case, pixel $L_2$ is a poor proxy for semantic uncertainty.

5. L.148 appears problematic. The architectural term is at most 1, while FID can be much larger depending on the feature extractor. So, the expression in L.148 collapses to the architectural difference.

6. L.273 seems inconsistent. The $U_{\text{prop}}$ term depends on $U_{\text{total}}$s (not on the $U_{\text{prop}}$s) via the weights. A form like $U_{\text{total}} = (1-\gamma)U_{\text{intrinsic}} + \gamma W U_{\text{total}}$ would be more consistent with the stated propagation.

7. In L.347, $D_w^{-1} W$ is not the normalized graph Laplacian. The normalized Laplacian is $I - D_w^{-1/2} W D_w^{-1/2}$.

8. The uncertainty maps in Fig. 2 are not clearly defined. They do not appear to be uncertainty in image space. Are they derived in a latent space? The surrounding discussion does not clearly connect to the maps.

**Questions:**

1. In L.181, what are the output categories when the model is a T2I generator?

2. Why does solving equation 24 lead to equation 25 ? Where does b come from?

3. The method targets uncertainty quantification. Could you explain what the quality metrics reported are suggesting, in this context? Why is LPIPS treated as higher is better? Also, sharpness seems flipped. larger gradient usually implies sharper, not lower.

4. In L.682, how were unambiguous, ambiguous, and OOD prompts defined and separated?

5. L.751 mentions a user study. Where are its design and results reported?

---

### Official Review · Reviewer_xBE5 · 2025-11-01

**Soundness:** 2
**Presentation:** 2
**Contribution:** 2
**Rating:** 2
**Confidence:** 4

**Summary:**

This paper presents a novel framework for uncertainty quantification (UQ) in text-to-image diffusion models.
The core proposal is to construct a graph where nodes represent different diffusion models and weighted edges represent their architectural and output similarity.
The paper introduces three strategies to propagate uncertainty across this graph: an iterative message-passing scheme (IPU), a spectral method leveraging the graph Laplacian for smoothness (SGUP), and a method that aggregates influence along all graph paths (PUI).
The authors provide extensive theoretical analysis covering convergence and complexity and report state-of-the-art performance on seven datasets, showing improvements over standard UQ baselines in both calibration and image quality metrics.

**Strengths:**

- Novel Conceptual Framework: The central idea of representing relationships between generative models as a graph to propagate and refine uncertainty is highly original and provides a new perspective on UQ beyond single-model or simple ensemble methods.
- Comprehensive Multi-Strategy Approach: The paper introduces three distinct and well-motivated propagation strategies (IPU, SGUP, PUI). This provides a flexible toolkit that can be adapted to different needs, such as iterative refinement, global smoothness, or detailed path-specific analysis.
- Strong Empirical Results: The experimental validation in Table 1 shows significant and consistent improvements over established baselines (Ensemble Sampling, MC Dropout, Bayesian Diffusion) across a wide range of UQ and image quality metrics (PICP, UCE, FID, LPIPS).
- Thorough Theoretical Analysis: The authors provide eight theorems with proofs covering the convergence, complexity, and smoothness properties of their methods. This attempt to ground the framework in solid theory is a commendable strength.

**Weaknesses:**

- Fundamental Mismatch between Methodology and Implementation:
The paper's core premise is critically undermined by a contradiction between the described methodology and the actual implementation.
Section 3.2 explicitly defines nodes as distinct diffusion models (e.g., DDPM, LDM).
However, the Appendix (A.3, lines 745-747) states that the experiments construct graphs over 'diffusion timesteps using cosine similarity of intermediate latent embeddings' for each sample.
This suggests the experiments were run on internal states of a single model, not a graph of different models. This is a severe flaw that makes the paper's central narrative misleading.
- Misalignment with Current Research Directions:
The paper frames the UQ problem as propagating uncertainty between different model architectures.
However, the key challenge in text-to-image UQ is widely considered to be semantic uncertainty arising from prompt ambiguity.
The work fails to compare against or even cite recent, highly relevant baselines like PUNC (Franchi et al., 2024), which directly addresses this problem and is a critical point of comparison.
- Technically Flawed Graph Construction:
The edge weighting formula in Equation (5) is mathematically problematic and poorly justified.
The formula simplifies to wij = min(S_arch, S_out), which is an arbitrary way to combine two similarity scores.
Furthermore, the theory presented in Section 4 (e.g., Theorem 1) relies on the weight matrix being row-stochastic, but the proposed edge weighting scheme does not guarantee this property, creating a disconnect between the theoretical claims and the method itself.
- Questionable Practical Utility:
The stated premise of requiring a graph of multiple, distinct diffusion models to perform UQ for a single generation is computationally prohibitive and practically infeasible for most real-world scenarios.
This raises serious questions about the utility of the proposed framework as described.
- Opaque Custom Metrics:
The paper's novel evaluation metrics, SGU-Score (Eq. 38) and PSUI (Eq. 39), are not clearly defined.
The definitions rely on unexplained terms like the operator FG and variables σu and σv, making it impossible to fully understand or trust the results reported using these metrics.

---

- General Limitations:
The paper's limitations section is well-considered.
However, it does not address how the graph construction process itself—the choice of models (or timesteps) to include and the specific similarity metrics used—could introduce its own biases, potentially leading to uncertainty estimates that are systematically skewed.

**Questions:**

- Could you please clarify the critical discrepancy between your methodology and implementation?
Was the graph constructed over different diffusion models as described in Section 3.2, or over the internal timesteps of a single model as suggested in Appendix A.3?
The validity of the paper's contribution hinges on this clarification.
- Can you provide a formal justification for the edge weighting formula in Equation (5), which resolves to $w_{ij} = min(S_{arch}, S_{out})$?
Why is this a meaningful way to combine architectural and output similarity, compared to other standard methods like a weighted average?
- How do you ensure that the weight matrix W meets the row-stochastic requirement for the convergence proof in Theorem 1?
The edge weighting scheme in Equation (5) does not inherently satisfy this property.
- Why was the recent and highly relevant baseline PUNC (Franchi et al., 2024), which was one of the first works to systematically address prompt-based uncertainty in T2I models, omitted from your experimental comparison?
- In the definitions for SGU-Score and PSUI (Eqs. 38-39), could you please formally define the Graph Fourier Transform operator FG and the variables σu and σv used in the PSUI formula?

---

### Note · Authors · 2025-11-23

I have read and agree with the venue's withdrawal policy on behalf of myself and my co-authors.